# Contribution of narrative therapy in reduction of anxiety, depression and PTSD among survivors of the genocide against the Tutsi in 1994 in Rwanda

**Emmanuel Mihigo Murengera**[1,2*], **Japhet Niyonsenga**[1,3*], **Eugene Rutembesa**[1], **Vincent Sezibera**[1], **Augustin Nshimiyimana**[1]

**1** Department of Clinical Psychology, College of Medicine and Heath Sciences, University of Rwanda, Kigali, Rwanda, **2** Department of Mental Health Nursing, School of Nursing Sciences, Mount Kigali University, Kigali, Rwanda, **3** Department of Humanities, Social Sciences and Cultural Industries, University of Parma, Parma, Emilia-Romagna, Italy

* mihigo91@gmail.com (EMM); niyonsengajaphet74@gmail.com (JN)

## Abstract

The 1994 genocide against the Tutsi in Rwanda caused profound psychological trauma among survivors, with long-lasting effects such as anxiety, depression, and post-traumatic stress disorder (PTSD). Narrative therapy, known for its ability to help individuals reconstruct their personal stories, is emerging as a promising intervention, though its effectiveness within Rwandan context has not been fully explored. This study therefore, aimed to assess the impact of narrative therapy on reducing symptoms of anxiety, depression, and PTSD in genocide survivors. It focused on how narrative therapy helps survivors reconstruct their narratives, enhance resilience, and foster emotional well-being. Eleven qualitative interviews were conducted with survivors of the 1994 Genocide against the Tutsi in Rwanda who participated in narrative therapy. Through purposive sampling, participants were selected for in-depth, semi-structured interviews. The interviews were transcribed and analyzed using thematic analysis to identify key themes and patterns. Thematic analysis revealed several key themes: survivors reported positive transformation in their narratives, which helped them reframe traumatic experiences and develop resilience. Participants strongly testified reduction in symptoms of anxiety, depression, and PTSD after engaging in narrative therapy. The communal aspects of therapy, such as group support, were also instrumental in fostering emotional recovery and a renewed sense of belonging. Narrative therapy appears to provide an effective framework for addressing psychological distress in genocide survivors by offering a structured process for narrative reconstruction and community healing. Further research is needed to confirm these findings and explore long-term outcomes. Culturally relevant mental health interventions, supported by sustainable funding, are essential to improving survivors' well-being and fostering resilience.

## 1. Introduction

The 1994 genocide against the Tutsi in Rwanda was one of the most horrific and devastating events of the 20th century, with over 800,000 people killed in just 100 days [1]. Many genocide

**Data availability statement:** All data can be found in the manuscript and Supporting Information files.

**Funding:** The authors received no specific funding for this work.

**Competing interests:** The authors have declared that no competing interests exist.

survivors continue to experience severe mental health disorders, particularly post-traumatic stress disorder (PTSD), depression, and anxiety [2,3]. The impact of these mental health conditions is both profound and persistent, affecting not only those who directly witnessed or survived the violence but also subsequent generations [4–7]. Addressing these psychological consequences has become a critical priority in Rwanda's mental health care system, particularly due to the prevalence of trauma-related disorders among the genocide survivors.

Research consistently highlights PTSD, depression, and anxiety as the most common mental health disorders affecting genocide survivors of the 1994 genocide against the Tutsi. In a national study, scholars reported that the most prevalent mental health conditions among Rwandans genocide survivors are major depressive episodes (35%), PTSD (27.9%), and panic disorder (26.8%) in Rwanda [8]. Similarly, recent study found that 37% of female survivors continued to suffer from PTSD [3]. These findings underscore the pressing need for therapeutic interventions to address the enduring impact of these mental health conditions.

Despite the high prevalence of these disorders, access to mental health care remains limited in Rwanda, particularly in rural areas where many survivors reside [9]. In response, narrative therapy has emerged as a promising intervention. Developed in the 1980s, narrative therapy is based on the idea that people make sense of their lives through the stories they tell about themselves [10]. This therapeutic approach encourages individuals to externalize their problems—viewing them as separate from themselves—and then to re-author their personal narratives in ways that emphasize resilience, empowerment, and healing [10]. Narrative therapy's focus on "meaning-making" is especially relevant in trauma recovery, where individuals often struggle to process their traumatic experiences and construct coherent narratives about their suffering [11].

In Rwanda, the cultural practice of oral storytelling plays a central role in both personal and collective identity formation. Narrative therapy aligns with these traditions by providing a structured way for survivors to reconstruct their life stories, emphasizing survival and strength [12]. By engaging with narrative therapy, survivors have the opportunity to reframe their experiences in ways that promote healing on both individual and communal levels, making it an especially suitable intervention in the Rwandan context.

Although research on narrative therapy in Rwanda is limited, existing studies highlight its potential. Authors have recently found that narrative therapy significantly improve resilience and reduce anxiety and ADHD in randomized controlled trials conducted among Rwandan orphan children [13,14]. However, much of the research on narrative therapy's effectiveness has been conducted outside of Rwanda. For instance, a meta-analysis by Lely et al [15]. found medium to large effect sizes for narrative therapy in treating PTSD and depression, enabling participants to reframe their traumatic experiences and regain a sense of agency. Scholars similarly reported its effectiveness in addressing various mental health issues in children, such as attachment disorders and family violence [16,17]. Studies involving war refugees [18] and other traumatized populations [19,20] also found that narrative therapy significantly reduced PTSD symptoms and helped trauma survivors reconstruct their identities.

Despite the demonstrated potential of narrative therapy in various populations, there is a significant research gap regarding its specific impact on adult Rwandan genocide survivors. Most studies [13,14] have focused on children and orphans, leaving limited evidence on how narrative therapy affects adults suffering from PTSD, depression, and anxiety. Moreover, while the meta-analysis by Lely et al [15]. supports narrative therapy's effectiveness in treating trauma, there is a lack of context-specific studies that consider Rwanda's unique cultural and historical background. Addressing this gap requires research that adapts narrative therapy to the specific needs of adult survivors, with a focus on long-term trauma effects and the potential for collective healing.

Given the high prevalence of PTSD, depression, and anxiety among genocide survivors and the strong cultural resonance of storytelling in Rwanda, this study seeks to fill this critical gap in the literature by examining the impact of narrative therapy on these three prevalent mental health conditions. By exploring how narrative therapy can facilitate individual and communal healing, this research aims to contribute to the development of culturally relevant mental health interventions in Rwanda, with important implications for mental health policies and practices in post-genocide settings.

## 2. Methodology

### 2.1 Study Design

An exploratory qualitative study design was used to investigate the impact of narrative therapy on reducing common mental disorders among survivors of the 1994 genocide against the Tutsi. This design was appropriate due to the sensitive nature of participants' experiences following narrative therapy sessions [21]. Semi-structured interviews provided a flexible framework for collecting rich qualitative data, allowing for an in-depth exploration of participants' thoughts, feelings, and beliefs regarding their therapy experiences.

### 2.2 Study Setting

The study was conducted in Rwanda, a low-income country in Sub-Saharan Africa, with an estimated population of 13.2 million engaged primarily in subsistence farming [22]. Rwanda is divided into four provinces and the City of Kigali, which are further organized into districts, sectors, cells, and villages. The research focused on survivors of the 1994 genocide against the Tutsi, estimated to number between 300,000 to 400,000 in the country [23]. The study, carried out from from July to November 2023, was conducted in partnership with Uyisenga Ni Manzi, an institution that utilizes narrative therapy in its interventions.

### 2.3 Sample and sampling method

The study recruited participants who were survivors of the 1994 genocide against the Tutsi and had completed multiple narrative therapy sessions. Purposive sampling was employed to select 11 genocide survivors who had undergone therapy sessions facilitated by Uyisenga Ni Manzi, ensuring a focused exploration of narrative therapy outcomes. The sample size was determined based on the principle of data saturation, which posits that data collection can cease when no new themes or information emerge [24]. The data saturation was achieved after the ninth interview, with 10[th] and 11[th] interviews conducted to confirm saturation had been reached. The inclusion criteria specified that participants must be genocide survivors who had attended a minimum of five narrative therapy sessions, with no restrictions based on gender or age.

### 2.4 Data collection and procedure

The process of data collection began with initial contact made via telephone. During these calls, the study was explained in detail, and participation was requested from potential participants. This outreach occurred one week prior to data collection to ensure that participants had ample time to prepare and confirm their willingness to take part in the study. All participants agreed to participate and were subsequently scheduled for face-to-face interviews at the Uyisenga Ni Manzi office, which was selected for its conducive environment for discussing sensitive personal experiences. Before the interviews commenced, informed consent was obtained from all participants.

Data were gathered using a semi-structured interview guide, which was developed by the authors and informed by relevant literature on trauma and narrative therapy [10,15,25,26]. The guide's questions were designed to explore the impact of narrative therapy on mental health outcomes such as PTSD, depression, and anxiety, while allowing flexibility for participants to express their personal stories. Sample questions included: "How would you describe the problem that has affected you, and what can you compare it to?", "Could you share your experiences participating in therapeutic sessions?", "Can you compare your situation before and after attending therapeutic sessions?" Probing questions, such as, "can you tell me what you felt about that?" Interviews lasted between 45 to 60 minutes and were audio-recorded with participant consent to ensure accuracy in data collection.

## 2.5 Data analysis

The qualitative data from interviews were audio-recorded, transcribed verbatim, and analyzed using a thematic analysis approach [27]. The analysis followed key steps, including familiarization with the data, generating initial codes, searching for, and refining themes. Initially, the transcripts were reviewed to gain a deep understanding of the participants' experiences with narrative therapy. The data were then coded to identify significant statements and patterns relevant to the research objectives. Themes were reviewed and refined for coherence and clarity. A final synthesis provided a comprehensive interpretation of the findings, illustrating how narrative therapy contributed to psychosocial recovery and the reduction of PTSD, depression, and anxiety symptoms. A team of three psychologists collaboratively analyzed the data, resolving discrepancies through discussion, with a Cohen's kappa score of 0.82, indicating substantial inter-rater agreement.

## 2.6 Ethical considerations

Given the sensitive nature of the research, ethical considerations were paramount. The study was approved by the Institution Review Board (IRB) of the college of Medicine and health sciences at the University of Rwanda (Ref: CMHS/IRB/162/2023). Participants were provided with detailed information about the study and gave their informed consent before participation. Confidentiality of participants was ensured by anonymizing participant data. Additionally, participants were informed that they could withdraw from the study at any time without consequences. Special care was taken to provide a supportive environment, and psychosocial support was available for participants who experienced distress during the interviews.

## 3. Results

### 3.1 Sociodemographic characteristics of the participants

Table 1 summarizes the demographic characteristics of 11 participants, ranging in age from 31 to 75 years old (M=49.5, SD=14.6), with the majority being female (7/11). Most participants have a primary education (7/11), two have secondary education and two are illiterate. Five participants are widows, reflecting the impact of historical events like the genocide. All participants are jobless, suggesting economic difficulties, and there is an even split between urban (5/11) and rural (6/11) residents. These characteristics provide important context for understanding how factors such as gender, education, and marital status may influence participants' mental health and engagement in narrative therapy.

### 3.2 Understanding the Multifaceted Experiences of survivors of the 1994 genocide against the Tutsi

Our analysis identifies ten major themes that capture the complex and multifaceted experiences of the genocide survivors, from their initial engagement with psychological care to

Table 1. Sociodemographic characteristics of the participants.

| Code | Age | Education | Sex | Marital Status | Occupation | Residence |
|---|---|---|---|---|---|---|
| MUH | 60 | Primary | Female | Widow(er) | Jobless | Rural |
| NYD | 57 | Illiterate | Female | Widow(er) | Jobless | Rural |
| UWI | 75 | Illiterate | Female | Widow(er) | Jobless | Rural |
| MUK | 62 | Primary | Female | Widow(er) | Jobless | Rural |
| MKT | 61 | Primary | Female | Widow(er) | Jobless | Rural |
| ESP | 36 | Secondary | Female | Married | Jobless | Urban |
| BON | 38 | Illiterate | Male | Married | Jobless | Urban |
| DEN | 37 | Primary | Female | Divorced | Jobless | Urban |
| EMU | 31 | Secondary | Male | Married | Jobless | Urban |
| MAGO | 35 | Primary | Female | Married | Jobless | Urban |
| VTT | 53 | Primary | Female | Separated | Jobless | Urban |

the therapeutic processes that helped them cope. These themes also highlight the challenges related to trauma, as well as the restorative impact of therapeutic interventions. Below, we present these themes, along with corresponding subthemes, frequencies, and verbatims to provide deeper insights into the survivors' lived experiences in Table 2.

**3.2.1. Initiation to Psychological care process.** Participants described diverse pathways through which they accessed psychological care, including self-initiation and external influences from community members. Local leaders, friends, and community health workers often played a critical role in encouraging survivors to seek therapeutic support. In several cases, the consultation was initiated by these community figures. For example, Participant **MUH** shared, "*The president of Ibuka was the one who called me and initiated the consultation.*" Another participant, **ESP,** mentioned "*It was through the Croix Rouge that I was invited for follow-up sessions after they noticed we had emotional crises at a commemoration event.*"

Other participants sought psychological support after recognizing the severity of their emotional and physical distress, marked by trauma symptoms such as insomnia, flashbacks, headaches, and bodily pain. As **MUK** recounted, "*I felt so weak and could barely move. I had constant headaches, and the memories of what happened kept coming back. That's when I knew I needed help.*" **UWI** reflected, "*A friend who was in the same group explained my problem and encouraged me to seek help. That's when I decided to consult.*"

**3.2.2. Reasons for search of psychological care.** Participants provided several reasons for seeking psychological care, primarily driven by intense emotional and physical distress that disrupted their daily lives. This distress often included symptoms of depression, anxiety, or PTSD. For many, these symptoms were exacerbated during genocide commemoration periods, where memories of past trauma resurfaced. **MUH** described how her experience: "*I had trauma-related symptoms like insomnia, confusion, and anger, especially during the commemoration period. I preferred staying at memorial sites because I felt relieved, but when I went home, I heard voices and couldn't sleep.*" Similarly, **VTT** shared, "*I experienced outbursts of anger and insomnia, and when I was asked too many questions, I would become silent and absent-minded.*"

For some, physical symptoms prompted them to seek care. Participant **NYD** explained, "*I felt hot, had severe headaches, and my neck felt like hot liquid was flowing down it. Those physical symptoms made me realize something was wrong.*" Others, like **ESP**, expressed fatigue and physical weakness, stating, "*I was always tired and didn't have the energy to do anything. That's when I realized I needed help.*" Many participants noted that social withdrawal pushed them to seek therapy. MKT shared, "*I had isolated myself from my children. I was angry all the*

**Table 2. Summary of themes and sub-themes.**

| Themes | Sub-themes | Frequency |
|---|---|---|
| **1. Initiation to Psychological Care Process** | | |
| | - Self-initiated | 4/11 |
| | - External figures (Ibuka president, friend, family) | 7/11 |
| | - Learned through community programs, word of mouth | 6/11 |
| **2. Reasons for Psychological Care** | | |
| | - Insomnia | 6/11 |
| | - Flashbacks | 5/11 |
| | - Feeling disconnected from family and community | 7/11 |
| | - Seeking connection with others | 7/11 |
| | - Physical symptoms (e.g., headaches, fatigue) | 6/11 |
| **3. Naming of Problems and Descriptions** | | |
| | - Described trauma using metaphors (e.g., hell, darkness) | 8/11 |
| | - Likened trauma to death or deep emotional pain | 7/11 |
| | - Difficulty finding words to describe emotional pain | 5/11 |
| **4. Experiences of Complex Trauma** | | |
| | - Recurrent memories of rape or multiple sexual assault | 5/11 |
| | - Continued psychological pain and suffering | 9/11 |
| | - Social rejection from community, neighbors, family | 6/11 |
| | -Re-traumatization during commemoration events | 6/11 |
| **5. Perceptions of Therapeutic Setting** | | |
| | - Found comfort in shared group experiences | 5/11 |
| | - Preferred individual therapy for deeper discussions | 6/11 |
| | - Safe space for emotional expression and breakdowns | 8/11 |
| | - Hesitancy to disclose sensitive issues in group therapy. | 4/11 |
| **6. Outcomes Following Narrative Therapy** | | |
| | - Better management of emotions, calming techniques | 9/11 |
| | - Re-established communication with family members | 6/11 |
| | - Restoring a Sense of Control over life | 7/11 |
| **7. Restored Trust Among Neighbors** | | |
| | - Slowly rebuilding trust in neighbours | 6/11 |
| | - Overcoming fear of neighbours implicated in violence | 5/11 |
| | - Developing empathy for neighbors' situations | 4/11 |
| | - Cautious trust, but hopeful for better relations | 5/11 |
| | - Instances of forgiveness between neighbors | 4/11 |
| **8. Instilling Hope** | | |
| | - Renewed ability to plan for the future | 7/11 |
| | - Found new sense of purpose through therapy | 5/11 |
| | - Expressed hope for better outcomes for children | 4/11 |
| | - Overcoming Suicidal Thoughts | 3/11 |
| | - Learning coping strategies for future challenges | 7/11 |
| **9. Sense of Family Belonging** | | |
| | - Rebuilt connections with family members | 5/11 |
| | - Found a new family within the therapy group | 6/11 |
| | - Mending Relationships with Children | 4/11 |
| | - Restored a sense of identity and belonging | 5/11 |

*(Continued)*

**Table 2.** (Continued)

| Themes | Sub-themes | Frequency |
|---|---|---|
| **10. What Truly Transpired During Therapy** | | |
| | - Healing through telling their stories | 8/11 |
| | - Guided reflection leading to breakthroughs | 6/11 |
| | - Learned practical techniques for managing emotions | 7/11 |
| | - Group activities helped rebuild trust | 5/11 |

*time and I didn't want to interact with anyone."* MUH similarly noted, *"I stopped talking to my neighbors and preferred staying alone at home."*

Participants also sought emotional relief and support, hoping to connect with others to alleviate their burdens. MKT said, *"I wanted to feel like I wasn't alone. That's why I went to the psychologist. I needed to talk to someone who could help me understand what I was going through."*

**3.2.3 Naming of problems and their descriptions.** Participants struggled to articulate their trauma, often relying on metaphors or spiritual references to convey the depth of their suffering. Some expressed an inability to fully verbalize their experiences, while others described their emotions as overwhelming and difficult to describe. MUH expressed her challenge: *"I cannot find a name for my problem. I just place it before God."* **MKT** similarly shared, *"something beyond me, something difficult to name."*

Many participants compared their trauma to death or extreme emotional pain. MUH stated, *"It feels like dying without really dying,"* while **VTT** described, *"It feels like being in hell, with the memories haunting me every day."* Participants frequently spoke of being over-whelmed by their trauma. DEN shared, *"It's like everything around me reminds me of what happened, and I can't escape it."*

**3.2.4 Experiences of complex trauma.** Participants provided detailed accounts of complex trauma stemming from the genocide, which resulted in persistent emotional and psychological distress. These accounts emphasized how deeply the trauma continued to affect their lives. MUH recalled her harrowing experience: *"I was raped, stabbed, and tortured. I lost my baby due to multiple rapes, and I have experienced physical and emotional complications ever since."* **VTT** shared a similar sentiment: *"The trauma I experienced has never left me. It feels like it's always with me, especially during certain times of the year."*

The emotional impact of trauma persisted long after the events, with participants reporting symptoms like emotional numbness, hypervigilance, and difficulty experiencing joy or connection. These long-term effects often left survivors feeling detached from their surroundings. One participant shared, *"I feel like I'm always on edge, always waiting for something bad to happen. I don't trust anyone, not even myself."* **ESP** added, *"I've distanced myself from everyone because I don't want anyone to see my pain."*

Many participants also faced social rejection as a result of their trauma. NYD explained, *"I was rejected by people in my community because of what I went through. I felt alone and aban-doned."* For several participants, annual commemoration events served as a painful reminder, often leading to re-traumatization. VTT reported, *"Every year during the commemoration period, it's like the trauma comes back. I see the bodies of my family members, and I can't stop the memories from flooding in."* VTT reported.

**3.2.5 Perceptions of therapeutic setting.** Participants had mixed perceptions of the therapeutic setting. Some found comfort in group therapy, describing it as a source of strength and solidarity. NYD shared, *"Group therapy gave me strength. It was comforting to know I wasn't alone."* However, others felt hesitant about sharing in a group. MKT explained, *"There*

*were things I didn't feel comfortable sharing in front of a group. Some memories are just too painful to talk about with everyone."*

Despite these differences, many participants appreciated the therapeutic setting as a safe space for emotional expression. VTT remarked, *"Therapy was the only place I felt safe enough to cry and let everything out."* Similarly, **ESP** noted, *"I needed that space where I could just be myself and not feel judged."*

**3.2.6 Outcomes following attendance at narrative therapy sessions.** Many participants reported positive outcomes from attending narrative therapy, particularly regarding emotional regulation and improved family relationships. MUH described how therapy helped her manage emotional triggers: *"Before therapy, I would break down every time I had a memory. Now, I have the tools to calm myself down and manage those feelings."* Similarly, **MUK** shared how therapy helped her reconnect with her family: *"Before, I distanced myself from my family because I didn't want them to see my pain. But therapy has helped me rebuild those connections."*

NYD reflected on the overall change in her emotional state, stating, *"I can now regulate my emotions. Before, I would feel overwhelmed by sadness, but now I can manage better, and my family has noticed the change."* Moreover, participants found therapy to be a critical tool in their journey toward healing. VTT shared that therapy helped her regain control over her emotions, saying *"Therapy gave me back a sense of control. I no longer feel like I'm at the mercy of my emotions."*

**3.2.7 Restored trust among neighbours.** A key outcome of therapy was the restoration of trust between survivors and their neighbours, especially in cases where neighbours had been involved in the genocide. Although this process took time, many participants spoke of efforts to reconcile and rebuild relationships. ESP described this process: *"Through the reconciliation programs, we started talking to our neighbours again. It's not easy, but it's a step forward."*

For others, reconciliation was linked to forgiveness. DE explained, *"I forgave the man who killed my brother. It was the hardest thing I've ever done, but it helped me let go of my anger."* Some participants also mentioned how therapy facilitated these reconciliation efforts. **VTT** shared, *"I don't trust everyone fully yet, but therapy gave me the strength to start trying."*

**3.2.8 Instilling hope.** Therapy helped instil a renewed sense of hope for many participants who had previously experienced hopelessness. After enduring long periods of despair, participants began to feel optimistic about the future. **MKT** noted, *"Before therapy, I didn't think there was any hope for me. Now, I feel like I can start planning for the future."* **ESP** added, *"Therapy gave me hope. I'm not afraid of what's ahead anymore."*

Participants also discussed how learning coping mechanisms during therapy contributed to their renewed hope. NYD shared, *"The coping strategies we learned in therapy have been a lifesaver. I feel more equipped to deal with difficult situations now."* DE expressed hope not only for herself but also for future generations, saying, *"I feel hopeful that my children won't have to suffer like I did. Therapy has given me tools to help them build a better future."*

Furthermore, for participants who had struggled with suicidal ideation, therapy helped them move beyond these dark thoughts. **MKT** noted, *"I used to think about ending my life all the time, but therapy helped me see that life is worth living."*

**3.2.9 Sense of family belonging.** Therapy facilitated the restoration of family connections and helped participants feel a renewed sense of belonging, both within their biological families and their therapeutic group. **MUH** reflected, *"Before, I distanced myself from my family because I didn't want them to see my pain. But therapy has helped me reconnect with them."* For some, group therapy became a substitute for the family they had lost. ESP explained, *"In the therapy group, we became like a family. We supported each other, and that made all the difference."*

**DEN** shared a similar sentiment: *"I never thought I'd find a family again, but this group has given me a sense of belonging."* Participants also discussed how therapy helped them repair

relationships with their children. MKT shared, *"Before therapy, I couldn't even talk to my children. Now we play together and talk about what's going on in our lives."*

Additionally, many participants shared how this renewed sense of belonging helped restore their sense of identity. VTT noted, *"I felt like I was nobody after everything that happened, but being part of this group has given me a new sense of who I am."*

**3.2.10  What truly transpired during therapy?.**  Participants reflected on the specific aspects of therapy that were most helpful to their healing, including storytelling, guided reflections, and emotional breakthroughs that allowed them to process their trauma in a safe environment. **DEN** shared, *"One of the exercises involved telling our stories, not just to relive the pain, but to reclaim our narratives. It helped me see my trauma in a new way."* Similarly, **NYD** added, *"The therapist taught us grounding exercises to calm our minds when we felt overwhelmed. It helped me feel more in control of my emotions."*

Emotional breakthroughs were significant moments in therapy, allowing participants to release years of suppressed grief, anger, and guilt. MUH shared,*"During one session, I finally talked about being raped during the genocide. I had kept it inside for so long. Once I let it out, I felt a huge weight lift off my shoulders."* ESP described a similar experience: *"There was one session where I couldn't stop crying. At first, I felt embarrassed, but the psychologist said it was okay. I hadn't allowed myself to feel that much in a long time, and afterward, I felt a kind of relief."*

Guided reflection exercises also helped participants process their trauma more deeply. ESP noted, *"The therapist guided me through my memories, and that helped me understand things I hadn't even realized before."* NYD similarly shared, *"There was a breakthrough moment in therapy when I finally understood that my trauma didn't define me."*

In addition to emotional breakthroughs, participants emphasized the importance of learning practical coping mechanisms. **VTT** explained, *"The exercises we did, like breathing and meditation, really helped me calm down during moments of stress."* **MKT** also reflected, *"I learned simple techniques, like holding my fingers and massaging them when I felt overwhelmed. It sounds small, but it made a big difference."*

Finally, group activities helped participants rebuild trust in others. **MUH** described, *"We did this activity where we exchanged gifts with other group members, and it made me realize that I can trust people again."* **VTT** added, *"The group games we played taught me how to interact with others and not feel afraid."*

## 4.  Discussion

This study explored the experiences of survivors of the 1994 genocide against the Tutsi in Rwanda, focusing on their initiation into psychological care and the outcomes of narrative therapy. The findings are particularly significant given Rwanda's historical context and the ongoing mental health challenges that genocide survivors face. Over 25 years after the genocide, many survivors continue to suffer from PTSD, depression, and anxiety [2,8]. This study contributes to the growing body of literature by providing an in-depth understanding of how narrative therapy, a culturally relevant intervention, impacts the mental health of adult genocide survivors.

The initiation of psychological care among survivors was largely influenced by community figures, which underscores the vital role of social networks in post-genocide Rwanda. Survivors often rely on community leaders or local organizations to facilitate their access to mental health services, particularly in rural areas where formal mental health infrastructure remains limited [9]. This aligns with findings by Rieder & Elbert [4] and extends the understanding of how physical symptoms, such as headaches, fatigue, and insomnia, often prompt survivors to seek psychological help. This finding is consistent with Van der Kolk's

work on trauma, which emphasizes the manifestation of trauma through both physical and emotional symptoms [28]. In Rwanda's context, where physical health often takes precedence, addressing these somatic symptoms in conjunction with psychological ones becomes imperative for holistic trauma care.

Survivors' reasons for seeking psychological help were primarily rooted in severe emotional distress, which was often exacerbated during annual genocide commemoration periods. The commemoration events serve as a trigger for the reactivation of traumatic memories, a pattern documented in other studies [29,30]. Our findings highlight the need for targeted mental health interventions during these emotionally charged times. Participants frequently described how physical manifestations of trauma, such as body pain and fatigue, were intertwined with their psychological distress, contributing to a cycle of suffering. This aspect of trauma, where physical and emotional symptoms are inseparable, has been extensively discussed in trauma literature, including Van der Kolk's work on the body's role in storing traumatic memories [28].

When it came to describing their trauma, many participants struggled to find the words to articulate their experiences. This difficulty in verbalizing trauma is a well-documented phenomenon in trauma studies [31]. Our study builds on this by showing how Rwandan survivors use culturally specific metaphors to describe their pain, such as likening their suffering to "dying without dying." This use of metaphor is particularly relevant in Rwanda, where oral storytelling is central to cultural identity [12]. The integration of narrative therapy, which relies on storytelling, aligns with these cultural practices, making it a particularly suitable intervention for Rwandan survivors. Future research should explore how the use of metaphors in therapeutic settings evolves over time and whether it aids in the processing of trauma.

Participants in this study also reported ongoing psychological distress related to their experiences of the genocide, particularly those who had endured extreme acts of violence, such as rape, torture, and the loss of loved ones. This is consistent with literature on complex trauma, which emphasizes the long-term psychological consequences of such experiences [32,33]. The emotional numbness, hypervigilance, and detachment described by participants echo the symptoms of PTSD, as documented in both international and Rwandan studies [34,35]. What our study adds is a deeper understanding of how these long-term trauma effects hinder survivors' reintegration into society and their ability to rebuild relationships. The isolation felt by survivors is not only social but emotional, emphasizing the importance of sustained psychological interventions that address deep-seated emotional wounds.

The mixed perceptions of the therapeutic setting, with some participants finding comfort in group therapy and others preferring individual sessions, highlight the need for flexible therapy models. While group therapy provides a sense of solidarity and shared experience, which is particularly valuable in post-conflict settings [36,37], some survivors feel hesitant to share sensitive issues in such settings. These findings underscore the importance of creating adaptable therapeutic environments that offer both group and individual therapy sessions, depending on the survivor's needs. Future research could investigate how survivors' preferences for therapeutic formats evolve over time and whether their comfort with certain formats increases as their healing progresses.

Narrative therapy was shown to have a positive impact on survivors, particularly in terms of emotional regulation and reconnection with family members. This aligns with research on the effectiveness of narrative therapy in treating trauma, as it enables survivors to reframe their experiences and regain a sense of control over their narratives [10,15,26,38]. The storytelling aspect of narrative therapy resonates deeply with Rwandan cultural traditions, where oral storytelling is a key part of identity formation [12]. This study adds to the existing literature by detailing the therapeutic techniques that survivors found most effective—such

as storytelling, guided reflections, and grounding exercises—showing the importance of these techniques in facilitating emotional breakthroughs and long-term healing.

One of the most profound findings of this study is the restoration of trust between survivors and their neighbours, including in cases where neighbours were involved in the genocide. This finding emphasizes the role of therapy in facilitating reconciliation, a topic explored in previous research [39]. Participants spoke of forgiveness and rebuilding relationships, showing how therapy can play a crucial role in community-level healing, not just individual recovery. Further research should investigate how these reconciliation efforts are sustained over time and how broader societal factors, such as government reconciliation programs, contribute to or hinder this process.

A renewed sense of hope was another key outcome of narrative therapy. This finding supports previous research on the role of hope in trauma recovery [40,41]. Survivors in this study felt more in control of their emotions and futures, with therapy helping them develop practical coping strategies. These outcomes align with studies that emphasize the role of narrative therapy in fostering resilience and empowerment [11,16]. The long-term focus of this study provides new insights into how survivors maintain hope and resilience in the face of ongoing challenges, highlighting the importance of continued mental health support in post-genocide contexts.

In addition to emotional healing, therapy helped survivors restore a sense of family belonging, whether with their biological families or through the therapeutic group. This finding aligns with previous research emphasizing the role of family connections in trauma recovery [42,43]. For many survivors, who lost family members during the genocide, the therapeutic group became a substitute family, providing much-needed emotional support. This study adds to the literature by showing how these therapeutic group relationships can provide long-term emotional support, creating new "families" for survivors who have lost their original ones.

Finally, participants reflected on the specific therapeutic techniques that were most beneficial to their healing, including storytelling, guided reflections, and emotional breakthroughs. These moments of emotional release were key in helping survivors process their trauma and move forward, which supports previous research on the effectiveness of narrative therapy in addressing trauma-related mental health issues [15]. Future research should explore how these techniques can be refined to better address the specific needs of Rwandan survivors, ensuring that therapeutic interventions remain culturally relevant and effective.

### Strengths and limitations

This study has several strengths. It provides a detailed, nuanced understanding of the long-term effects of trauma and the role of narrative therapy in the healing process for genocide survivors. The study also emphasizes the importance of culturally sensitive interventions, offering insights into how narrative therapy aligns with Rwanda's oral storytelling traditions. However, the study has limitations. The reliance on self-reported data may introduce recall bias, as participants might have difficulty accurately remembering or might selectively report their experiences. Additionally, the cross-sectional design limits the ability to assess long-term changes in mental health outcomes. Furthermore, the study focuses on adult survivors, leaving a gap in understanding how younger generations affected by the genocide may respond to similar interventions. Finally, as qualitative study it focused on a specific geographic region, which may limit the generalizability of the findings to other survivor populations across Rwanda.

## 5. Conclusion

The analysis of the participant interviews highlights the complex emotional and psychological journeys of survivors of the 1994 genocide against the Tutsi, underscoring the critical role that

psychological care, particularly narrative therapy, in the healing process. Providing safe spaces for survivors to share their experiences, process their emotions, and reconnect with families and communities has proven effective in alleviating symptoms of anxiety, depression, and PTSD. Therapeutic techniques such as storytelling, guided reflection, and group support facilitated emotional breakthroughs, restore family bonds, and offered survivors renewed hope. The communal aspects of therapy were particularly powerful, as many participants reported a renewed sense of belonging and trust within their communities, highlighting the importance of incorporating social and community-based healing approaches in trauma recovery.

Expanding psychological care programs that integrate narrative therapy and community-based healing across Rwanda is essential to increase survivor access. Mental health professionals should be trained in culturally appropriate narrative therapy and group interventions to enhance therapeutic effectiveness. Policymakers must prioritize sustainable, long-term funding for trauma recovery programs to ensure continuity of care. Additionally, further comparative studies with larger sample sizes are recommended to comprehensively assess the therapeutic efficacy of narrative therapy and explore its integration with other approaches for greater impact on emotional regulation, family reconciliation, and community reintegration.

## Acknowledgments

I express gratitude to the Uyisenga Ni Manzi organization for their invaluable support in this work. Special thanks to the two psychologists who provided guidance and assistance during the in-depth interviews.

## Author contributions

**Conceptualization:** Emmanuel Mihigo M..

**Formal analysis:** Eugene Rutembesa, Japhet Niyonsenga.

**Methodology:** Emmanuel Mihigo M., Eugene Rutembesa, Augustin Nshimiyimana, Japhet Niyonsenga.

**Resources:** Eugene Rutembesa, Vincent Sezibera.

**Supervision:** Eugene Rutembesa, Vincent Sezibera, Augustin Nshimiyimana.

**Visualization:** Augustin Nshimiyimana.

**Writing – original draft:** Emmanuel Mihigo M..

**Writing – review & editing:** Vincent Sezibera, Augustin Nshimiyimana, Japhet Niyonsenga.

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
