## [Decision Letter · Decision Letter 0]

17 Jul 2024

PMEN-D-24-00207

Contribution of Narrative Therapy in Reduction of Anxiety, Depression and PTSD among Survivors of the Genocide against the Tutsi in 1994 in Rwanda

PLOS Mental Health

Dear Dr. Mihigo Murengera,

Thank you for submitting your manuscript to PLOS Mental Health. After careful consideration, we feel that it has merit but does not fully meet PLOS Mental Health’s publication criteria as it currently stands. Therefore, we invite you to submit a revised version of the manuscript that addresses the points raised during the review process.

The manuscript has been evaluated by two reviewers, and their comments are available below.

Reviewer 2 has raised a number of major concerns which span the entire manuscript. These include overhauling the literature review, providing much more detail on the methods, considering expanding the analysis, and revising the discussion. Addressing these concerns represents a substantial amount of work. Given the extent of the issues identified, I considered issuing a reject decision. However, I think it is possible to conduct a extensive revision of your manuscript (should you choose to), so I am issuing a major revision decision instead.

Could you please carefully revise the manuscript to address all comments raised?

We look forward to receiving your revised manuscript.

Kind regards,

Steve Zimmerman, PhD

PLOS Staff Editor

Journal Requirements:

Additional Editor Comments (if provided):

Reviewers' comments:

Reviewer's Responses to Questions

**Comments to the Author**

1. Does this manuscript meet PLOS Mental Health’s publication criteria ? Is the manuscript technically sound, and do the data support the conclusions? The manuscript must describe methodologically and ethically rigorous research with conclusions that are appropriately drawn based on the data presented.

Reviewer #1: Yes

Reviewer #2: Partly

2. Has the statistical analysis been performed appropriately and rigorously?

Reviewer #1: N/A

Reviewer #2: N/A

3. Have the authors made all data underlying the findings in their manuscript fully available (please refer to the Data Availability Statement at the start of the manuscript PDF file)?

Reviewer #1: Yes

Reviewer #2: No

4. Is the manuscript presented in an intelligible fashion and written in standard English?

Reviewer #1: Yes

Reviewer #2: No

5. Review Comments to the Author

Reviewer #1: Source 8: The publication year should be added.

Source 10: This source should be removed due to its outdated nature.

Source 15: This source should also be removed as it is more than 10 years old.

Please, correct them

Reviewer #2: Dear Authors,

Thank you for manuscript- The subject of the article is very important and there is no doubt that exposure to stressful events has a long-term impact on a person's mental health and Narrative therapy may be one of the treatments for the psychopathology developed.

The manuscript examined the effectiveness of Narrative Therapy in Reducting psychopathology developed including Anxiety, Depression and PTSD among Survivors of the Genocide against the Tutsi in 1994 in Rwanda.

I have a few comments for your consideration:

1. The authors should clearly state the objectives, methods, results and conclusion with suggestions in the abstract which is unclear and can aid the readers.

2. Introduction is very week only focusing on the nature of narrative therapy while citing a few studies from the contexts other than the current study’s context. It is recommended to first provide the background of the Tutsi genocide while contextualizing the terms in local including genocide, trauma/violence, mental health, mental disorders so on and so forth. The authors should cite the relevant literature from other parts of the world. Additionally, theoretical framework and clear rational should be provided before the objectives. The term broad objective should be used instead of general objective and the authors could come up with a few specific objectives to have more clear understanding of the purpose of the study to the readers. The nature of psycho pathological symptoms developed should also be provided by citing the literature from Rwanda resulting from trauma and violence. Why authors have focused only on the anxiety, depression and PTSD only?

3. The interview schedule was "self constructed", meaning it was developed by the authors? If so, was the interview schedule informed by the literature or just questions that the authors came up with? Please justify.

4. The 10 themes were derived from the interviews conducted; the authors should examine whether they could be further categorized into sub-themes. Moreover, the themes should be further framed within a framework or model that functionally speaks to adjustment of the participants. The authors are suggested to write the themes more coherently and provide numbering to the themes by providing the verbatim responses while explaining the themes to aid the readers.

5. The biggest drawback of the paper is that the findings have not been discussed therefore, it is recommended to adequately discuss the findings while citing relevant literature. It is also recommended to provide separate sections for limitations, implication and/or suggestions which are the essence of this kind of study from a specific context. This has made the paper very week.

I think this is a very interesting manuscript but it needs more clarification and information completion with respect to introduction and methods. Results need to be revised and findings to be discussed adequately while citing the appropriate literature. Implications and limitations to be included.

Thank you and good luck

6. PLOS authors have the option to publish the peer review history of their article (what does this mean? ). If published, this will include your full peer review and any attached files.

**Do you want your identity to be public for this peer review?** For information about this choice, including consent withdrawal, please see our Privacy Policy .

Reviewer #1: **Yes: ** Tamara Shusterman

Reviewer #2: **Yes: ** Dr. Aehsan Ahmad Dar

---

## [Decision Letter · Decision Letter 1]

13 Nov 2024

PMEN-D-24-00207R1

Contribution of Narrative Therapy in Reduction of Anxiety, Depression and PTSD among Survivors of the Genocide against the Tutsi in 1994 in Rwanda

PLOS Mental Health

Dear Dr. Mihigo M.,

Thank you for submitting your manuscript to PLOS Mental Health. After careful consideration, we feel that it has merit but does not fully meet PLOS Mental Health’s publication criteria as it currently stands. Therefore, we invite you to submit a revised version of the manuscript that addresses the points raised during the review process.

We look forward to receiving your revised manuscript.

Kind regards,

Yasodha Maheshi Rohanachandra, MBBS, MD (Psychiatry)

Academic Editor

PLOS Mental Health

Journal Requirements:

Additional Editor Comments (if provided):

Reviewers' comments:

Reviewer's Responses to Questions

**Comments to the Author**

1. If the authors have adequately addressed your comments raised in a previous round of review and you feel that this manuscript is now acceptable for publication, you may indicate that here to bypass the “Comments to the Author” section, enter your conflict of interest statement in the “Confidential to Editor” section, and submit your "Accept" recommendation.

Reviewer #1: All comments have been addressed

2. Does this manuscript meet PLOS Mental Health’s publication criteria ? Is the manuscript technically sound, and do the data support the conclusions? The manuscript must describe methodologically and ethically rigorous research with conclusions that are appropriately drawn based on the data presented.

Reviewer #1: Yes

3. Has the statistical analysis been performed appropriately and rigorously?

Reviewer #1: Yes

4. Have the authors made all data underlying the findings in their manuscript fully available (please refer to the Data Availability Statement at the start of the manuscript PDF file)?

Reviewer #1: Yes

5. Is the manuscript presented in an intelligible fashion and written in standard English?

Reviewer #1: Yes

6. Review Comments to the Author

Reviewer #1: A significant amount of work has been done to address the reviewers' comments. In my opinion, the article in its current version is ready for publication in the journal

7. PLOS authors have the option to publish the peer review history of their article (what does this mean? ). If published, this will include your full peer review and any attached files.

**Do you want your identity to be public for this peer review?** For information about this choice, including consent withdrawal, please see our Privacy Policy .

Reviewer #1: **Yes: ** Tamara Shusterman

Reviewer #2

Thank you for manuscript- The subject of the article is very important and there is no doubt that exposure to stressful events has a long-term impact on a person's mental health and Narrative therapy may be one of the treatments for the psychopathology developed.

The manuscript examined the effectiveness of Narrative Therapy in Reducting psychopathology developed including Anxiety, Depression and PTSD among Survivors of the Genocide against the Tutsi in 1994 in Rwanda

I have a few comments for your consideration:

1. The authors should clearly state the objectives, methods, results and conclusion with suggestions in the abstract which is unclear and can aid the readers.

2. Introduction is very week only focusing on the nature of narrative therapy while citing a few studies from the contexts other than the current study’s context. It is recommended to first provide the background of the Tutsi genocide while contextualizing the terms in local including genocide, trauma/violence, mental health, mental disorders so on and so forth. The authors should cite the relevant literature from other parts of the world. Additionally, theoretical framework and clear rational should be provided before the objectives. The term broad objective should be used instead of general objective and the authors could come up with a few specific objectives to have more clear understanding of the purpose of the study to the readers. The nature of psycho pathological symptoms developed should also be provided by citing the literature from Rwanda resulting from trauma and violence. Why authors have focused only on the anxiety, depression and PTSD only?

3. The interview schedule was "self constructed", meaning it was developed by the authors? If so, was the interview schedule informed by the literature or just questions that the authors came up with? Please justify.

4. The 10 themes were derived from the interviews conducted; the authors should examine whether they could be further categorized into sub-themes. Moreover, the themes should be further framed within a framework or model that functionally speaks to adjustment of the participants. The authors are suggested to write the themes more coherently and provide numbering to the themes by providing the verbatim responses while explaining the themes to aid the readers.

5. The biggest drawback of the paper is that the findings have not been discussed therefore, it is recommended to adequately discuss the findings while citing relevant literature. It is also recommended to provide separate sections for limitations, implication and/or suggestions which are the essence of this kind of study from a specific context. This has made the paper very week.

I think this is a very interesting manuscript, but it needs more clarification and information completion with respect to introduction and methods. Results need to be revised and findings to be discussed adequately while citing the appropriate literature. Implications and limitations to be included.

---

## [Editor Report · Decision Letter 2]

26 Nov 2024

PMEN-D-24-00207R2

Contribution of Narrative Therapy in Reduction of Anxiety, Depression and PTSD among Survivors of the Genocide against the Tutsi in 1994 in Rwanda

PLOS Mental Health

Dear Dr. Mihigo M.,

Thank you for submitting your manuscript to PLOS Mental Health. After careful consideration, we feel that it has merit but does not fully meet PLOS Mental Health’s publication criteria as it currently stands. Therefore, we invite you to submit a revised version of the manuscript that addresses the points raised during the review process.

EDITOR:

The authors have addressed the comments and made substantial revisions to the manuscript, which has greatly enhanced the quality of the manuscript.

I have one suggestion to improve the manuscript. In the section on data collection and procedure, the authors state that " Data were gathered using a semi-structured interview guide, which was developed by the authors and informed by relevant literature on trauma and narrative therapy". Can the authors please cite the articles that they referred to when they developed the interview guide.

We look forward to receiving your revised manuscript.

Kind regards,

Yasodha Maheshi Rohanachandra, MBBS, MD (Psychiatry)

Academic Editor

PLOS Mental Health
---

## [Editor Report · Decision Letter 3]

26 Feb 2025

Contribution of Narrative Therapy in Reduction of Anxiety, Depression and PTSD among Survivors of the Genocide against the Tutsi in 1994 in Rwanda

PMEN-D-24-00207R3

Dear Mr Mihigo M.,

We are pleased to inform you that your manuscript 'Contribution of Narrative Therapy in Reduction of Anxiety, Depression and PTSD among Survivors of the Genocide against the Tutsi in 1994 in Rwanda' has been provisionally accepted for publication in PLOS Mental Health.

Best regards,

Karli Montague-Cardoso

Executive Editor

PLOS Mental Health